# Multivariate Assessment of Vaccine Equity in Cambodia: A Longitudinal VERSE Tool Case Study Using Demographic and Health Survey 2004, 2010, and 2014

**DOI:** 10.3390/vaccines11040795

**Published:** 2023-04-04

**Authors:** Yijin Zhao, Joshua Mak, Gatien de Broucker, Bryan Patenaude

**Affiliations:** 1International Vaccine Access Center, Johns Hopkins University, Baltimore, MD 21231, USA; 2Department of International Health, Johns Hopkins University Bloomberg School of Public Health, Baltimore, MD 21205, USA

**Keywords:** vaccines, equity, monitoring and evaluation, Cambodia, national immunization program, concentration index, absolute equity gap

## Abstract

Cambodia has exhibited great progress in achieving high coverage in nationally recommended immunizations. As vaccination program managers plan interventions to reach last-mile children, it is important to consider issues of equity immunization priority setting. In this analysis, we apply the VERSE Equity Tool to Cambodia’s Demographic and Health Survey for the years 2004, 2010, and 2014 to evaluate multivariate equity in vaccine coverage for 11 vaccination statuses, emphasizing the results of the 2014 survey for MCV1, DTP3, fully immunized for age (FULL), and zero dose (ZERO). The largest drivers of vaccination inequity are socioeconomic status and the educational attainment of the child’s mother. MCV1, DTP3, and FULL exhibit increasing levels of both coverage and equity with increasing survey years. The national composite Wagstaff concentration index values from the 2014 survey for DTP3, MCV1, ZERO, and FULL are 0.089, 0.068, 0.573, and 0.087, respectively. The difference in vaccination status coverage between the most and least advantaged quintiles of Cambodia’s population, using multivariate ranking criteria, is 23.5% for DTP3, 19.5% for MCV1, 9.1% for ZERO, and 30.3% for FULL. By utilizing these VERSE Equity Tool outputs, immunization program leaders in Cambodia can identify subnational regions in need of targeted interventions.

## 1. Introduction

Immunizations are one of the most cost-effective public health interventions, averting millions of cases of illness and premature death annually [1]. However, vaccine access is not universal. In 2018, more than 13 million children aged below one do not receive a single dose of vaccination globally [2]. Furthermore, the COVID-19 pandemic’s strain on health delivery systems has resulted in delays and declines in routine immunization worldwide with 25 million children missing out on certain vaccinations in 2021 [1,3]. The Immunization Agenda 2030 (IA2030) calls for the reduction in deaths caused by vaccine-preventable diseases and increased delivery of comprehensive and universal vaccinations, with an emphasis on coverage and equity. Timely evaluation of vaccine coverage and equity will be valuable for identifying effective programs and policies for achieving the goals set out by IA2030 [4].

As for many low- and middle-income countries, Cambodia’s immunization program faces challenges. The National Immunization Program Strategic Plan 2008–2015 declared that health facilities need to facilitate vaccination access to hard-to-reach populations at greater risk of under-vaccination. However, major accessibility barriers include poor health education due to insufficient governmental financial support for establishing village-based social mobilization. Moreover, difficulties in maintaining sufficient financial support for outreach services has resulted in low service delivery quality, and the persistently low wages in the public sector have negatively impacted the performance of human resources [5]. Additionally, the utilization of public health sectors for outpatient services is comparably low [6,7]. In 2014, the private health sector accounted for 61% of health service provision, whereas that of the informal sector accounted for 26% [7]. In response, Cambodia’s Ministry of Health emphasized a more homogenous distribution of the health workforce at all levels of the health system, more accountability of local health centers, and improvements on communication links between communities and health centers [8].

Despite these ongoing challenges, Cambodia’s national Demographic and Health Surveys (DHSs) have illustrated stable improvements in vaccine coverage over time. For example, the percentage of children who had never received a single national immunization program (NIP) vaccine for which they were eligible was 7%, 5%, and 3% in 2004, 2010, and 2014, respectively. Although high coverage attainment is observed nationally, regional disparities stand out in the Eastern region, especially in Mondulkiri and Ratanakiri districts. For example, the 2014 DHS reports national BCG’s coverage rate as 94%, whereas that of Mondulkiri and Ratanakiri is 79.1%. To combat geographic heterogeneity, the Ministry of Health has proposed improvements to immunization service quality and subnational micro planning for its NIP [8].

The World Health Organization (WHO) defines health inequities as “differences in health status or in the distribution of health resources between different population groups, arising from the social conditions in which people are born, grow, live, work and age” [9]. Additionally, inequities are often shaped multidimensionally by social determinants such as education attainment, districts, and enrollment in health insurance coverage. Thus, health inequities could arise from the uneven distribution of these social conditions, both in ways associated with and independent from socioeconomic status [10]. To avoid unfair equity, resources should be distributed in such a way that different groups have equitable performances on different health evaluations [11]. In 2008, the Commission on Social Determinants of Health (established by WHO) appealed to the governments to take action to resolve the avoidable health inequities that are attributed to the “the circumstances in which people grow, live, work, and age, and the systems put in place to deal with illness,” for the welfare of everyone. However, health inequities are not influenced by individuals’ decisions alone. From a broader perspective, economic, social, and political policies also affect social conditions which can translate to desirable health outcomes [12].

Many conventional measurements of health inequities base disadvantage on the wealth quintile alone [13,14]. Recently, USAID developed an equity tool with univariate evaluation of equity that is restricted to the wealth factor only [15]. However, evaluating health inequities with any univariate approach will likely lead to an incomplete and misleading picture of true inequity. Other researchers have explored the use of factors other than measures of socioeconomic status for univariate assessments of equity [16,17]. As part of the Vaccine Economics Research for Sustainability and Equity (VERSE) project, the VERSE Equity Toolkit was developed to address the need to assess equity in health outcomes from a multidimensional equity lens. Building upon the prior equity literature, this toolkit enables users to evaluate equity multidimensionally by calculating a composite metric for ranking individuals, by determining the extent of inequity, and allowing for insights that would be missed by evaluating metrics derived from socioeconomic status ranking alone [18].

In this paper, we apply the VERSE composite equity assessment tool [18] to 11 pediatric vaccination outcomes in Cambodia’s 2004, 2010, and 2014 Demographic and Health Surveys (DHS). Our analysis focuses on the equity results for four vaccination statuses for the most recent available DHS survey year of 2014: zero-dose (a child who never received a single dose of vaccine on Cambodia’s national immunization schedule; ZERO), full immunization (a child receiving all vaccines they are eligible for according to their age; FULL), measles-containing vaccine first dose (MCV1), and diphtheria tetanus toxoid and pertussis vaccine third dose (DTP3). Zero-dose status is a priority for Gavi, the Vaccine Alliance as there remain challenges in children not receiving the latter doses of their immunization schedule [19]. This also leads to questions of equity in children receiving all vaccines that they are eligible for according to age. Given Cambodia’s success in controlling measles, despite outbreaks in other countries in Southeast Asia, it is interesting to look at MCV1 vaccination equity as an indicator of health system performance. Additionally, including the results of DTP3 allows us to better understand the factors associated with children not receiving the last doses according to their immunization schedules. We then decompose the observed inequities to derive the relative correlated inequity over time both at the national and subnational levels. Results from this study can allow decisionmakers to monitor the progress on equitable achievement of coverage targets and examine heterogeneity in equitable achievement across regions.

## 2. Materials and Methods

This analysis relies on the equity tool produced under the Vaccine Economics Research for Sustainability & Equity (VERSE) project to evaluate vaccine equity across three Cambodia DHS surveys [18]. We emphasize that the VERSE Equity Tool is meant to be used as a measurement metric to enable monitoring and evaluation teams to track equity in health interventions over time, as well as to help inform policy action by enabling decisionmakers to identify subnational regions with particularly low vaccination coverage and/or equity levels.

The VERSE methodology combines the socioeconomic equity works of Wagstaff and Erreygers with models for direct unfairness in health outcomes rooted in social choice theory, a theoretical framework accounting for preferences and context to make a collective decision and applied to the topics of health care inequity [20,21,22,23,24,25,26]. In accordance with said frameworks, we must define fair and unfair factors that contribute to inequity in vaccination outcomes. Fair factors refer to those deemed acceptable in determining a child’s vaccination status. In this analysis, our fair factor is whether the child is of appropriate age to receive the vaccine based on the national immunization schedule. Unfair factors are those which, in an ideal world, should not influence health resource access. For our purposes, unfair factors are the sex of the child, whether the child lives in a rural or urban area, district of residence, household insurance status, education level of the mother, and wealth quintile.

The principal output of the VERSE equity tool is a concentration index (CI) for childhood vaccination status. In equity analyses, a CI illustrates the extent of inequity in an outcome of interest derived from how the outcome is distributed across a group of individuals who experience different levels of advantage. The VERSE Equity Tool produces two types of CI. The Wagstaff CI is more influenced by individual observations residing at the extremes of disadvantage status and is the version of the CI referenced throughout the results of this paper [20]. Meanwhile, the Erreygers Corrected CI adjusts for the mean of the dependent variable. Though we do not report the Erreygers Corrected CI values in the main text, we provide them in the Appendix A.

Whereas traditional CIs rank observations relative to wealth or socioeconomic status, evidence suggests that using socioeconomic status or wealth alone as a proxy for advantage, with respect to healthcare utilization, is insufficient for describing the complex systems that contribute to accessing childhood vaccines. The intersectionality theory suggests that multiple social characteristics of individuals may result in compounding effects on health inequities and should thus be considered for health equity evaluation under the context of social dynamics such as caste and wealth status [27,28]. The VERSE equity tool expands upon these traditional models by ranking observations using a composite propensity for relative disadvantage metric calculated using multiple unfair drivers of inequity, including the traditional measures of socioeconomic status (wealth quintile). After computing the concentration index, a Kitagawa–Blinder–Oaxaca decomposition is performed to estimate the association of each unfair correlate of vaccination status with overall vaccination inequity [21,29]. To highlight the limitations of one-dimensional assessments of equity, we report univariate wealth-only CI values and contrast them with composite index CI values, which rank observations according to our multi-variate metric for disadvantage.

Another VERSE output is the absolute equity gap (AEG), which can be described as the gap in vaccine coverage or status prevalence between the top and bottom quintiles of individuals ranked by relative composite disadvantage. The AEG provides one method for presenting the inequity observed in the data in real units, relatable to policy targets and natural coverage units (percentages) rather than in abstract numeric terms.

Data for this analysis are sourced from the Cambodia’s DHS for the years 2004, 2010, and 2014 to evaluate trends in vaccination equity over multiple surveys [30]. Vaccines from the NIP available in the DHS for all survey years include measles conjugate vaccine dose 1 (MCV1), diphtheria-pertussis-tetanus doses 1 to 3 (DTP1-3), oral polio vaccine 1 to 3 (OPV1-3), and measles conjugate vaccine dose 1 (MCV1). We also consider indicators for three comprehensive vaccination statuses. Zero-dose status (ZERO) is defined as a child under 12 months of age never receiving a single dose of NIP vaccine. Full immunization status (FULL) refers to a child under 24 months of age receiving all NIP vaccines that they are eligible for at their age. Complete immunization status (COMPLETE) indicates that a child that is over 24 months of age has received all NIP vaccinations. As a note, MCV1 is excluded when defining FULL due to its common delivery via supplemental immunization activity (SIA) catch-up campaigns, capturing a wider range of ages relative to the national schedule age.

## 3. Results

In this section, we provide the equity results for ZERO, FULL, MCV1, and DTP3 vaccination statues. Results for other vaccines and vaccination statuses as well as survey years 2004 and 2010 can be referenced in the Appendix A. When discussing subnational results, some districts are paired (as indicated by the ”&” symbol) as a result of the Cambodia DHS data structure.

### 3.1. Zero-Dose

At the national level, the zero-dose prevalence rate is 3.29%. The wealth-based Wagstaff CI is −0.014, whereas its composite counterpart is 0.573 (Table 1). If we consider Wagstaff CIs ranking individuals by wealth quintile alone, the Wagstaff CI is close to zero, indicating little inequity in zero-dose status. However, the composite Wagstaff CI is 0.573 (95% confidence interval: 0.501–0.645), which is significantly higher than that of other vaccination statuses. The AEG shows that vaccination coverage amongst the most disadvantaged quintile of zero-dose children would need to increase by 9.1 percentage points to achieve similar levels of zero-dose prevalence as the most advantaged quintile. Wealth quintile is the greatest unfair contributor to zero-dose status, which accounts for 35.4% of inequity (Figure 1). The second greatest unfair contributor is maternal education level, to which 34.9% of inequity is attributed. In comparison, district, whether the household resides in an urban or rural area, sex of the child, and health insurance coverage status are responsible for 7.3%, 1.5%, 2.1%, and 0.4% of inequity, respectively. As the sole fair contributor to vaccination status, being underage for vaccination is responsible for 9.2% of status variation. Unexplained variation accounts for 9.2% of inequity, suggesting that the model explains approximately 90% of variation in overall observed inequity.

A total of 3 of the 19 districts have a zero-dose prevalence rate greater than 5%: Kampot & Kep, Mondulkiri & Ratanakiri, and Pursat (Table 2; Figure 2). Moreover, 10 out of 19 districts report zero-dose prevalence rates of less than 3%. In particular, Phnom Penh has a zero-dose prevalence rate of 0%. On the other hand, Banteay Meanchey has the highest composite Wagstaff CI of 0.839, followed by Battambang & Pailin at 0.789, and Kampong Chhnang at 0.727, all of which indicate high levels of zero-dose status inequity (Table 3; Figure 3). In total, 12 of 19 districts have composite Wagstaff CIs of less than 0.5. Though Mondulkiri & Ratanakiri and Kratie have high zero-dose prevalence rates, their Wagstaff CIs are 0.592 and 0.334, respectively. Meanwhile, Banteay Mean Chey and Battambang & Pailin have very low zero-dose prevalence rates but high composite Wagstaff CI values of 0.839 and 0.789, respectively.

### 3.2. Fully-Immunized for Age

The percentage of children in Cambodia fully immunized for their age is 78.4%. The national composite Wagstaff CI is 0.077 (0.068–0.086). Its wealth-based counterpart is 0.031 (0.014–0.048) (Table 1). Maternal education level is the most influential unfair contributor to full immunization status, accounting for 31.9% of variation (Figure 1). Wealth quintile also plays a significant role, responsible for 20.9% of inequity. The AEG shows that the rate of full immunization needs to be increased by 30.3 percentage points for the most disadvantaged quintile to have prevalence levels like those of the most advantaged quintile. Furthermore, full-immunization status has the largest AEG out of all statuses, which is unsurprising given that it envelops all vaccines except for MCV1. Since 37.9% of inequity is attributed to unexplained variation, inclusion of additional variables may improve the ability to explain the drivers of inequity (Figure 1).

All of Cambodia’s districts have full-immunization coverage rates greater than 50%, ranging from 51.9% for Mondulkiri & Rattanakiri to 89.3% for Banteay Meanchey. A total of 17 out of 19 districts have full-immunization rates above 70% (Table 2; Figure 2). Mondulkiri & Rattanakiri, Kampot & Kep, and Kratie experience the greatest levels of relative inequity with composite Wagstaff CIs of 0.235, 0.11, and 0.109, respectively (Table 3; Figure 3).

### 3.3. MCV1

National MCV1 coverage is estimated to be approximately 80.3% with a composite Wagstaff CI of 0.047 (0.038–0.056) and a wealth-based Wagstaff CI of 0.024 (0.011–0.037) (Table 1). The AEG indicates that the national MCV1 coverage rate needs to rise by 19.5% for the most disadvantaged quintile to achieve a coverage rate like that of the most advantaged quintile. Wealth quintile is the most influential unfair contributor, accounting for 52.2% of inequity (Figure 1). Maternal education accounts for 19.7% of inequity, while district-level difference is responsible for 0.3%. Urban/rural designation, sex of child, and health insurance status do not appear to be strong drivers of MCV1 vaccination variation. Unexplained variations account for 16.1% of inequity.

A total of 10 out of 19 districts have a MCV1 coverage rate of over 80%. Mondulkiri & Rattanakiri is the district with the lowest MCV1 coverage rate (66.8%) (Table 2; Figure 2). Overall, inequity in MCV1 vaccination among districts is low, though Mondulkiri & Rattanakiri are outliers with a composite Wagstaff CI of 0.143 (Table 3; Figure 3).

### 3.4. DTP3

National DTP3 coverage is estimated to be 82% with a wealth-based Wagstaff CI of 0.031 (0.017–0.045) and a composite Wagstaff CI of 0.062 (0.055–0.069) (Table 1). There is a 23.5% DTP3 coverage gap between the most disadvantaged and most advantaged quintiles. Maternal education and wealth quintile are the most influential factors owing to status inequity, accounting for 41.7% and 38.1%, respectively (Figure 1). Other unfair drivers included in the model do not appear to have substantial impact on status variation. Additionally, unexplained variation accounts for only 6.7% of inequity. Mondulkiri & Rattanakiri, Kratie, and Kampot & Kep experience the greatest relative inequity per the Wagstaff CI (Table 3; Figure 3).

### 3.5. Trends over Time

Across the different surveys years, we observe that the zero-dose prevalence rate has decreased from 7% (year 2004) to 5% (year 2010), and then to 3% (year 2014), though the equity level (1–Wagstaff CI) has also decreased from 0.57 (year 2004) to 0.427 (year 2014), suggesting a more inequitable distribution in the zero-dose vaccination status and indicating that those remaining zero-dose patients remain among the most disadvantaged across the multiple criteria used in the ranking of composite disadvantage (Figure 4). On the other hand, MCV1 prevalence rate has increased from 69% to 80%. Although there is only a minor increase in its equity levels from 0.903 in 2004 to 0.905 in 2010, the equity level then increased significantly and reached 0.953 in 2014. Referring to full-immunization and DTP3 vaccination status, both coverage rates and equity levels have stable improvements over the survey years (Figure 5).

## 4. Discussion

For Cambodia’s 2014 DHS, the most influential contributors to inequity for DTP3, MCV1, zero-dose status, and full immunization status are maternal education level and wealth quintile. Notably, the model explains the most variation in zero-dose, MCV1 and DTP3 immunization status, with much higher levels of unexplained variation attributable to full-immunization vaccination status. This finding suggests that further investigation into the barriers to access for birth dose vaccines, as well the causal pathway to completing all immunizations, is warranted to improve the targeting of immunization programs. In addition, the decomposition of inequity shows that district, whether individuals reside in an urban or rural area, the sex of the child, and health insurance coverage status contribute minimally to inequity after accounting for education and socio-economic status.

Generally, zero-dose prevalence rates are low in all districts except in Mondulkiri & Ratanakiri, where the prevalence is 16.5%. Though inequity in zero-dose status exists in most districts, it is especially high in the districts of Banteay Meanchey (composite Wagstaff CI: 0.839), Battambang & Pailin (0.789), and Kampong Chhnang (0.727), which also possess some of the lowest zero-dose prevalence rates. This indicates that zero-dose status in these districts is heavily correlated with the observable measures of disadvantage, whereas in some of the higher prevalence districts, it is more randomly distributed across levels of disadvantage. Furthermore, we observe that greater zero-dose prevalence is not necessarily associated with greater inequity. Representative examples to illustrate this would be Banteay Meanchey and Battambang & Pailin, districts that have very low zero-dose prevalence rates but very high inequity. On the other hand, it suggests that districts that have relatively lower performance in one metric may perform better in the other, such as in the case of Mondulkiri & Ratanakiri with a relative lower inequity in zero-dose status, despite its extremely high zero-dose prevalence rate. In situations where low zero-dose prevalence is associated with high inequity, the interpretation should be that the remaining zero-dose individuals in those settings are among the most vulnerable and disadvantaged, whereas in settings with similar prevalence and lower inequity, the zero-dose children are less concentrated amongst the most vulnerable and disadvantaged, which may indicate other structural barriers to access in those settings. Additionally, the coverage rates of full-immunization, MCV1, and DTP3 vaccination status tend to be greater in western Cambodia. Most districts have moderately low composite Wagstaff CIs for these vaccination statuses, suggesting high equity levels. Strengthened vaccination programs are needed in Mondukiri & Katanakiri as it stands out significantly for underperformance in both measures of coverage and equity.

Cambodia has a relatively homogenous coverage map with moderately high coverage rates for most vaccinations, with the notable exception of Mondulkiri & Ratanakiri. In comparison with many other low- and middle-income countries, it also has higher DTP3 prevalence rates [31,32]. For DTP and POLIO, both vaccines administered in multiple doses, we witness a decrease in coverage and an increase in composite inequity with subsequent doses, further highlighting the need for decisionmakers to continue bolstering outreach attempts to vaccinate children with the last routine doses. Although the DHS surveys indicate that the DTP3 coverage rate in 2014 was 82%, the National Immunization Program (NIP), reports a prevalence of 98% for 2015 [8]. This highlights differences in the methodologies employed by different surveys, which can result in different outcomes under the same analysis tool.

Greater equity in childhood vaccination may be associated with improved maternal education. According to Cambodia: Demographic and Health Survey 2014, in most districts, approximately half of the women have attained some level of primary education; however, only 5.6% of the women in total have completed primary education [33]. The USAID/Cambodia Gender Assessment published in 2010 suggested that wealth quintile and maternal education level, two of the most predominant unfair contributors in our case, are closely tied. Households with low socioeconomic status are more likely to give up their daughters’ education rather than their sons’ due to a lack of recourses and out of need for labor force [34]. Moreover, improvements in girls’ upper educational attainment are not evenly distributed across all wealth quintiles, as higher income quintiles experience more benefits [34,35,36]. Decisionmakers should account for different factors associated with maternal education levels to improve women’s access to education, which may help facilitate more equitable vaccination. Studies have indicated that antenatal care services that involve health education have positive impacts on maternal health knowledge and should also be encouraged [37,38]; therefore, similar strategies may be considered with vaccination.

Non- and under-vaccination may also be related to various factors not considered in this analysis, such as patient waiting times, media use, proximity to health centers, migration, ethnic minorities, antenatal care services, and mother’s mobility, etc. Residents of Cambodia complain of lengthy waiting times in public hospitals [39]. This may cause people, especially those of lower socioeconomic status, to avoid visiting these hospitals for vaccinations. Furthermore, Cambodia has a strong tradition of self-medication via alternative medicine sources and there is evidence that effective utilization of media can help instill public trust in vaccines in lower- and middle-income settings [39,40,41,42]. There is also a positive correlation between educational attainment and media use. Among the women who have not received any education, 58.2% of them do not access newspapers, television, or radio [33]. This could suggest that there is a self-reinforcing cycle of information for women of higher educational level through media use to attain more knowledge and thus improve their education level.

In addition, the regional differences in maternal educational level are observed to have certain correlations with their performances in immunization status. Mondulkiri & Ratanakiri has the highest percentage of women receiving no education (34.5%) and with no literacy skills (46.9%). In contrast, Phenom Penh has the lowest percentages, of 4.1% and 8.6%, respectively. Additionally, the teenage pregnancy rate in Modulkiri & Ratanakiri is especially high with 33.8% of women aged between 15–19 having started childbearing [33]. Once a girl starts pregnancy, their education may be suspended or even terminated. High early pregnancy rates could hinder efforts to improve the maternal educational level in the region. Promoting family-planning services (including contraceptive services) should be encouraged to help prevent teenage pregnancy from disrupting primary education.

This analysis is subject to several limitations. Though the selected unfair drivers of inequity explain variation well for several vaccination statuses, the share owed to unexplained sources of variation is still substantial, indicating that there are additional factors contributing substantially to the observed inequity for certain vaccines. While the exclusion of such factors does not change the composite metric or degree of the observed inequity in the study, it does limit the ability to describe the correlated of observed inequality. The current iteration of the VERSE equity tool only decomposes vaccine inequity at a national level; therefore, care needs to be taken in attributing these inferences at the district level. At its core, the VERSE equity tool is meant to be a descriptive and evaluative tool and not used to derive causal inferences. As such, it is not recommended that decisionmakers utilize the decomposition outputs to target correlate-specific interventions, but rather utilize the coverage and equity values to identify underperforming districts and to monitor progress toward reducing inequities over time. The utilization of DHS data is also subject to biases caused by self-report or proxy report (e.g., maternal retrospective memories concerning children’s health status) [43,44].

In this work, we have shown the decomposition of inequity levels for zero-dose, full-immunization, MCV1, and DTP3 status attributed to various social determinants. Maternal education and socio-economic status of households are observed to be the determinant unfair contributors for these immunization statuses, whereas other social characteristics included in the decomposition account minimally for inequity. The coverage rates and equity levels in Cambodia are generally high; however, geographical inequities persist, suggesting the need for more targeted and reinforced immunization programs in those regions. Furthermore, we have observed maternal education and wealth quintile to be strong contributors to variability in childhood vaccination status. Though wealth quintile is a long-examined factor in the equity space, quantification of the role of maternal education in health resource access presents another avenue for interventions aimed at health equity. Through the VERSE Equity Tool CI outputs, we also stress the limitations of univariate measurements of equity. Our analysis shows that wealth-only CI values are frequently different from composite index CI values, revealing that reliance on one-dimensional approaches is inadequate.

Though Cambodia has made great progress in childhood immunization, inequities persist. The identification of the challenges that vaccination programs face, combined with regular program evaluations of key metrics, such as those provided by the VERSE equity tool, can help decisionmakers better plan routine immunization programs and investments to improve outreach to disadvantaged populations.

## Figures and Tables

**Figure 1 vaccines-11-00795-f001:**
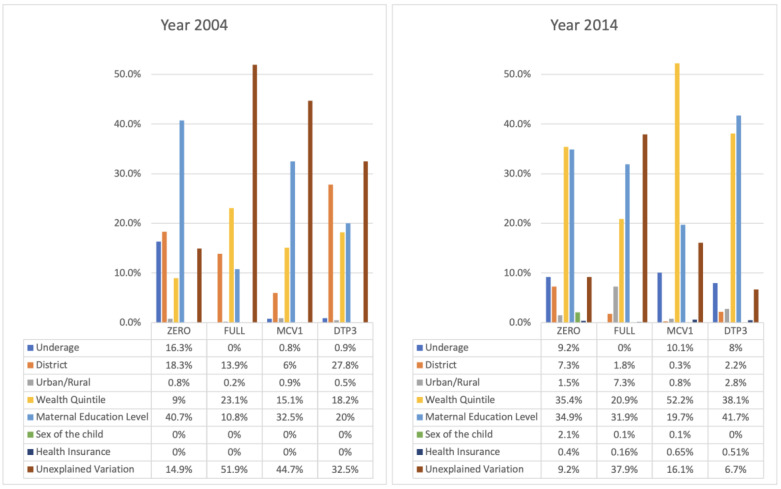
Share of variation in ZERO, FULL, MCV1, and DTP3 statuses associated with fair and unfair drivers for DHS year 2004 (**left**) and 2014 (**right**).

**Figure 2 vaccines-11-00795-f002:**
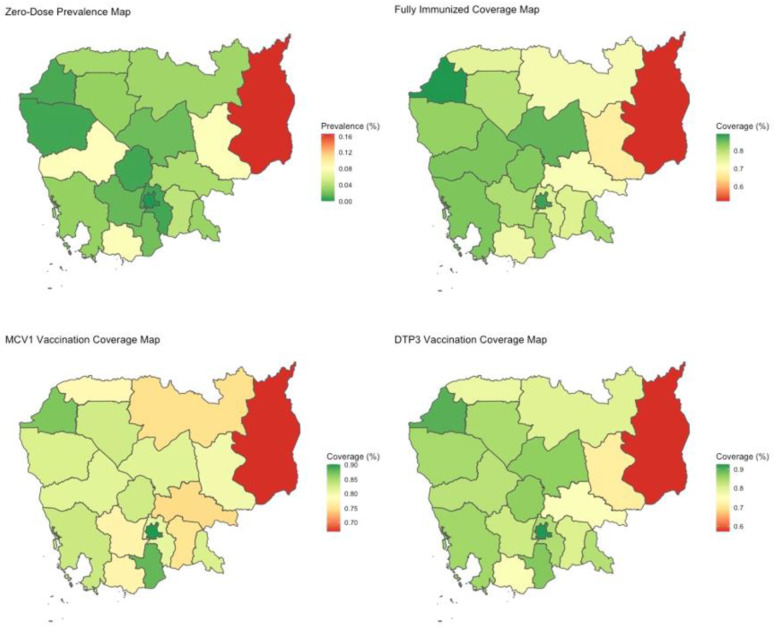
Heat maps portraying coverage/prevalence for ZERO, FULL, MCV1, and DTP3 for survey year 2014. Spatial divides represent individual or pairs of Cambodia’s administrative districts.

**Figure 3 vaccines-11-00795-f003:**
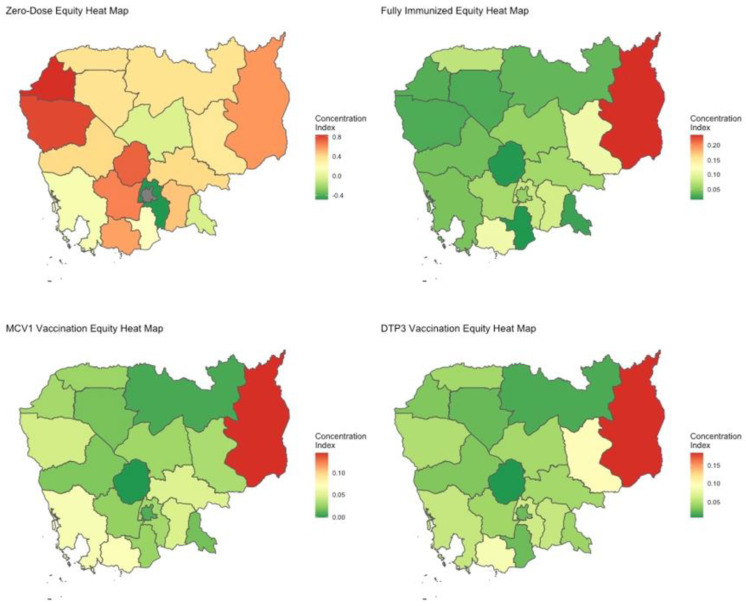
Heat maps of relative extent of equity, using the composite Wagstaff CI, for ZERO, FULL, MCV1, and DTP3 for survey year 2014. Spatial divides represent individual or pairs of Cambodia’s administrative districts.

**Figure 4 vaccines-11-00795-f004:**
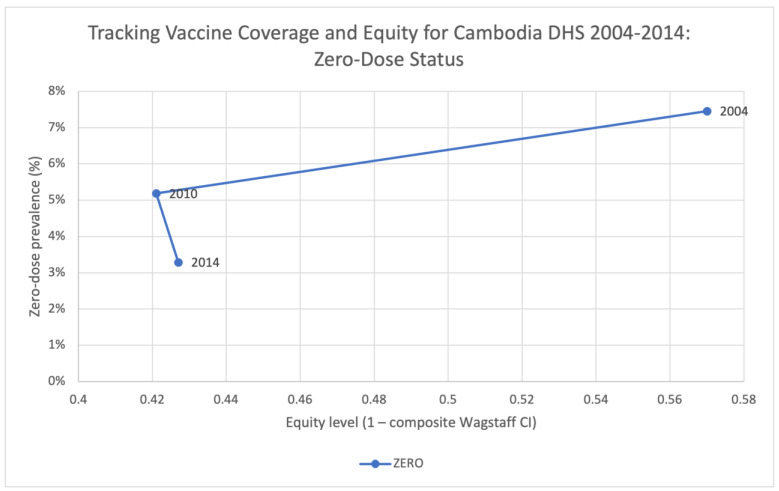
Tracking vaccine coverage and equity for Cambodia DHS 2004–2014: ZERO.

**Figure 5 vaccines-11-00795-f005:**
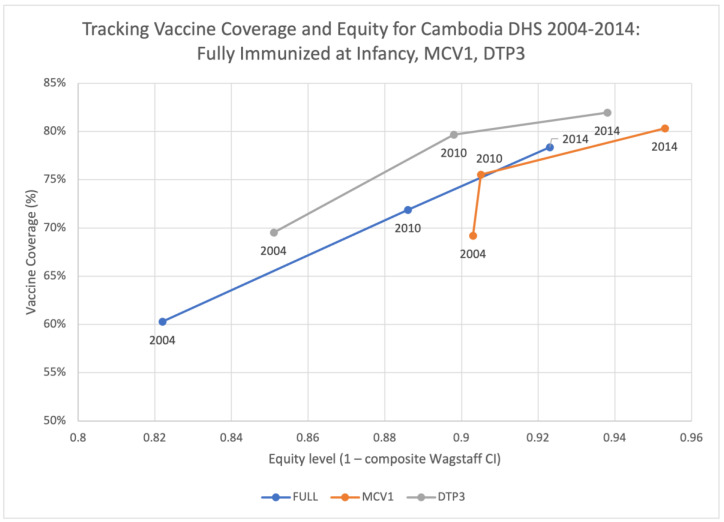
Tracking vaccine coverage and equity for Cambodia DHS 2004–2014: FULL, MCV1, DTP3.

**Table 1 vaccines-11-00795-t001:** National-level metrics for coverage/prevalence, Wagstaff CIs using wealth-only and composite ranking metrics, and composite-based AEGs for all Cambodia national immunization program vaccines and comprehensive vaccination statuses for survey year 2014.

		Wagstaff Concentration Index		
Vaccine or Outcome ^1^	Proportion Covered	Wealth Only ^2^	Composite	Absolute Equity Gap
BCG	94.0%	0.013	0.031	0.146
DTP1	91.8%	0.016	0.033	0.155
DTP2	87.4%	0.026	0.05	0.206
DTP3	82.0%	0.031	0.062	0.235
OPV1	92.5%	0.014	0.029	0.122
OPV2	87.6%	0.024	0.048	0.197
OPV3	81.6%	0.03	0.063	0.239
MCV1	80.3%	0.024	0.047	0.195
ZERO	3.3%	−0.014	0.573	0.091
FULL	78.4%	0.031	0.077	0.303
COMPLETE	84.5%	0.028	0.067	0.28

Notes: ^1^ ZERO, the child had not received any vaccine by 12 months old; FULL, the child is under 24 months old and is fully immunized for their age; COMPLETE, the child is over 24 months and completed the routine pediatric immunization schedule. ^2^ Concentration index based on households ranked by socioeconomic status (as defined in the DHS) only.

**Table 2 vaccines-11-00795-t002:** Coverage/prevalence for individual districts and district pairs for national immunization program vaccines and vaccination statuses for survey year 2014.

District	Vaccine or Health Outcome Coverage										
	BCG	DTP1	DTP2	DTP3	OPV1	OPV2	OPV3	MCV1	ZERO	FULL	COMPLETE
Banteay Meanchey	96.8%	94.9%	91.8%	90.0%	95.7%	92.6%	90.2%	86.9%	0.8%	89.3%	97.2%
Battambang & Pailin	99.5%	95.2%	89.0%	85.0%	95.3%	89.3%	83.4%	82.1%	0.7%	82.7%	87.2%
Kampong Cham	92.0%	85.1%	81.6%	75.0%	88.6%	82.9%	76.1%	74.4%	3.8%	71.7%	76.8%
Kampong Chhnang	98.9%	95.5%	90.6%	87.0%	95.5%	90.6%	86.5%	82.8%	0.7%	83.9%	92.7%
Kampong Speu	95.7%	92.7%	89.5%	82.0%	93.0%	88.8%	81.7%	77.0%	1.6%	80.5%	86.9%
Kampong Thom	96.3%	93.2%	90.0%	87.0%	94.1%	90.9%	86.6%	81.5%	1.8%	86.0%	96.4%
Kampot & Kep	87.8%	86.4%	80.0%	76.0%	87.4%	80.6%	75.2%	77.1%	7.8%	73.2%	78.3%
Kandal	96.3%	96.6%	92.0%	84.0%	95.6%	90.5%	81.9%	81.0%	0.7%	76.0%	85.1%
Kratie	84.4%	87.8%	78.8%	71.0%	87.7%	79.4%	71.3%	79.9%	8.0%	67.0%	75.3%
Mondulkiri & Ratanakiri	79.1%	76.7%	67.4%	57.0%	78.8%	67.2%	56.8%	66.8%	16.5%	51.9%	59.6%
Oddar Meanchey	93.8%	88.8%	81.8%	78.0%	89.5%	81.6%	77.4%	77.6%	3.5%	75.0%	79.6%
Phnom Penh	97.6%	98.3%	95.5%	92.0%	97.6%	95.7%	91.2%	90.1%	0.0%	88.1%	93.2%
Preah Sihanouk & Koh Kong	94.6%	92.6%	88.8%	85.0%	92.4%	88.6%	85.4%	83.1%	2.9%	84.2%	89.8%
Preah Vihear & Stung Treng	90.1%	92.3%	87.8%	80.0%	92.8%	87.7%	79.0%	74.7%	3.3%	72.4%	80.4%
Prey Veng	92.7%	91.5%	86.4%	80.0%	91.8%	86.4%	80.3%	75.2%	4.5%	76.1%	87.6%
Pursat	92.8%	91.8%	88.0%	83.0%	92.5%	88.8%	83.4%	81.4%	8.1%	84.3%	86.6%
Siem Reap	95.0%	95.9%	92.0%	84.0%	95.2%	92.5%	84.9%	82.8%	2.9%	79.8%	80.2%
Svay Rieng	93.5%	90.2%	86.9%	84.0%	90.1%	86.9%	83.6%	82.2%	3.1%	81.0%	88.1%
Takeo	95.5%	92.8%	88.9%	87.0%	94.4%	88.9%	85.7%	87.9%	1.9%	80.9%	87.0%

**Table 3 vaccines-11-00795-t003:** Composite Wagstaff concentration indices for individual districts and district pairs for national immunization program vaccines and vaccination statuses for survey year 2014.

District	Vaccine or Health Outcome Wagstaff Composite Concentration Index										
	BCG	DTP1	DTP2	DTP3	OPV1	OPV2	OPV3	MCV1	ZERO	FULL	COMPLETE
Banteay Meanchey	0.019	0.017	0.037	0.031	0.017	0.038	0.032	0.03	0.839	0.029	0.013
Battambang & Pailin	−0.003	0.019	0.043	0.049	0.015	0.031	0.046	0.045	0.789	0.027	0.049
Kampong Cham	0.022	0.035	0.032	0.043	0.022	0.032	0.045	0.05	0.406	0.06	0.033
Kampong Chhnang	−0.004	−0.001	0	0.006	0.004	0.002	0.014	−0.001	0.727	0.015	0.011
Kampong Speu	0.016	0.02	0.029	0.042	0.019	0.028	0.042	0.024	0.646	0.062	0.046
Kampong Thom	0.015	0.022	0.031	0.044	0.02	0.027	0.04	0.028	−0.032	0.053	0.013
Kampot & Kep	0.032	0.059	0.067	0.088	0.052	0.065	0.093	0.068	0.563	0.11	0.094
Kandal	0.002	0.014	0.03	0.061	0.014	0.029	0.065	0.03	−0.492	0.08	0.065
Kratie	0.049	0.034	0.064	0.099	0.037	0.067	0.095	0.032	0.334	0.109	0.089
Mondulkiri & Ratanakiri	0.111	0.1	0.148	0.184	0.099	0.159	0.191	0.143	0.592	0.235	0.221
Oddar Meanchey	0.018	0.023	0.037	0.043	0.022	0.04	0.046	0.027	0.363	0.074	0.064
Phnom Penh	0.001	0	0.012	0.025	0.003	0.014	0.023	0.008	NA	0.053	0.031
Preah Sihanouk & Koh Kong	0.02	0.031	0.05	0.057	0.032	0.048	0.057	0.065	0.114	0.042	0.032
Preah Vihear & Stung Treng	0.015	0.01	0.011	0.016	0.007	0.004	0.018	0.006	0.367	0.034	0.019
Prey Veng	0.005	0.015	0.035	0.057	0.016	0.04	0.059	0.048	0.471	0.085	0.05
Pursat	0.016	0.005	0.019	0.031	0.007	0.016	0.029	0.019	0.407	0.04	0.039
Siem Reap	−0.002	0.004	0.015	0.024	0.004	0.017	0.012	0.017	0.363	0.027	0.045
Svay Rieng	0.003	0.02	0.037	0.041	0.021	0.035	0.035	0.016	−0.042	0.021	0.019
Takeo	0.019	0.031	0.028	0.025	0.01	0.013	0.026	0.026	0.163	0.014	−0.013

## Data Availability

All data used in this analysis are publicly available through the Demographic and Health Survey website.

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
