# Peer review of "Multivariate Assessment of Vaccine Equity in Cambodia: A Longitudinal VERSE Tool Case Study Using Demographic and Health Survey 2004, 2010, and 2014"

_vaccines, 2023, doi:10.3390/vaccines11040795_

Round 1

Reviewer 1 Report

In-text citations throughout the paper need to be on the left of the period, not the right.

i.e. “Vaccines are good [3].” And not “Vaccines are good. [3]”

43: grammar

44: under-vaccination

70: quotes

89: “VERSE” hasn’t yet been defined in the paper, assuming it is an acronym

97: focuses

99: Gavi is undefined

10: deconstruct or isolate might be a better term than decompose

Results: overall the text is great and very informative. No major issues.

The figure text could be improved, adding at the very least some of the statistical analyses and more description. I get that a lot of the figures speak for themselves, but you need more text. Your figure text shouldn’t just be the same text as the figure title (Figures 4, 5).

Overall the paper is good and gets your story across. A few small things but nothing major

Reviewer 2 Report

This manuscript was described about the relationship between multivariate vaccine equity and vaccine coverage between districts in Cambodia by a group of Johns Hopkins University. They concluded the largest drivers of vaccination inequity were socioeconomic status and the educational attainment of the child’s mother, and they presented Cambodia had exhibited great progress in achieving high coverage from 2004 to 2014. I agree with their conclusion. My comments are following;

Major comments

Table1, 3 and 4: Authors were used 2 types of concentration index, Wagstaff and Erreygers. However, authors did not explain the difference between Wagstaff and Erreygers. Also, when different results between Wagstaff and Erreygers were obtained, authors did not discuss the reasons. Authors should clarify the differences between Wagstaff and Erreygers and should discuss the obtained results.

Figure 4 and 5: This manuscript was described the results of survey 2014 detailly. In 2004 and 2010, vaccine coverage and equity level only were presented in Figure 4 and 5. I understood improvement both vaccine coverage and equity level in Cambodia from 2004 to 2014. However, I would like to know what kind of drivers were eliminated in vaccine inequity between 2004 and 2014 or were still same vaccine inequity in 2014.

Minor comments

L51: “reponse” will be misspelling.

L63: “combate” will be misspelling.

L95: The meaning of “VERSE” should be explained in L95, although it was explained in L113 and 114. Abbreviations should be defined the first time they appear.

L98 and 99: The meaning of “MCV1” and “DTP3” should be explained in 98 and 99, although it was explained in L161 and 162. Abbreviations should be defined the first time they appear.

Table 2, 3 and 4: I do not know any detail about Cambodia. Therefore, I could not fully understand Table 2, 3 and 4, because I did not know the characteristics of each district, where was urban, rural or others. Please present the characteristics of each district.

Round 2

Reviewer 2 Report

I confirmed your corrected manuscript. I could understand results simply, because results analyzed by Wagstaff CI only were presented. In the text, “Wagstaff CI” and “CI” without Wagstaff were mixed. Please make sure whether those were intended.

I have no additional comments.

Author Response

Thank you very much for taking the time to review our paper and for your thoughtful in comments and suggestions. Based on your final comment, we have modified the text such that we are consistent in referencing the Wagstaff CI throughout the main body of the manuscript. Best wishes.